# Nanopore Sequencing for T-Cell Receptor Rearrangement Analysis in Cutaneous T-Cell Lymphoma

**DOI:** 10.3390/cancers16213700

**Published:** 2024-11-01

**Authors:** Cassandra Cieslak, Carsten Hain, Christian Rückert-Reed, Tobias Busche, Levin Joe Klages, Katrin Schaper-Gerhardt, Ralf Gutzmer, Jörn Kalinowski, Rudolf Stadler

**Affiliations:** 1Department of Dermatology, Johannes Wesling Medical Centre, University Hospitals of the Ruhr-University of Bochum (UKRUB), University of Bochum, 32429 Minden, Germany; 2Medical School OWL, Bielefeld University, 33594 Bielefeld, Germany

**Keywords:** cutaneous T-cell lymphoma, Mycosis fungoides, nanopore sequencing, T-cell receptor rearrangement analysis

## Abstract

Precise analysis of T-cell receptor clonality is essential in lymphoma diagnostics, particularly for cutaneous T-cell lymphomas like Mycosis fungoides (MF) and Sézary Syndrome (SS). The current gold standard, based on next generation sequencing (NGS), is precise and accurate but time- and cost-intensive. This study established a workflow for the analysis of the T-cell receptor repertoire using third generation nanopore sequencing. Results are similarly accurate to NGS for cutaneous T-cell lymphoma, but less time-consuming and cost-intensive.

## 1. Introduction

Cutaneous T-cell lymphoma (CTCL), in particular Mycosis fungoides (MF), has a very good prognosis in the early stages IA to IIA [1,2,3] which deteriorates dramatically in advanced stages [4]. While early diagnosis is crucial, the diagnosis is often delayed; according to a retrospective analysis, diagnosis is made on average three years after the onset of clinical symptoms [5,6,7,8]. In 20% of CTCL patients, however, the disease progresses, which leads to reduced life expectancy and poorer quality of life [6]. Early diagnosis is based on the clinical presentation, histological and immunohistology criteria, as well as on the detection of the rearrangement of the TCR in the skin, which includes both T-cell receptor gamma (TRG) and T-cell receptor beta (TRB) genes [9,10]. The detection of blood involvement additionally requires the recording of defined blood parameters measured by FACS analysis, while clonality analysis in TRB can be also included [11,12,13].

The methods for clonality analysis from the skin have changed in recent years, especially with regard to accuracy, time, and costs. Beginning with Southern hybridization for alleles [14] and evolving to fragment-length detection of highly multiplexed PCR products, the current gold standard method is NGS-based readout of TRG and TRB amplicon sequences. This development was significantly propelled by the international consortium BIOMED (now named EuroClonality), which devised multiplex primer sets for TCR and immunoglobulin genes. These standardized protocols have since become widely adopted in laboratories and clinics [15]. The main workflow of the Euro clonality protocol contains DNA extraction, one targeted PCR amplification (e.g., TRG), a second PCR for the patient barcoding, and serval purification steps to achieve a clean sample with a specified fragment length. The PCR products have usually been analyzed using fragment-based detection methods like polyacrylamide gel electrophoresis or capillary gel electrophoresis (CE).

Although fragment analysis by multiplex primer PCR and subsequent CE are still routinely used for TCR analysis, their results only provide information about the fragment length, without any sequence information. Pitfalls of this methods are false-positive pseudo clonality results, where a diverse repertoire of TCR sequences of similar fragment length are interpreted as one clonal TCR. In addition, the limit of detection of fragment-based detection is low [16], leading to false negative evaluation, especially in early stages. In detail, the evaluation of various studies in recent years shows a sensitivity of 44–90% for diagnostic evaluation made by PCR-electrophoresis, compared to sequencing-based methods with a sensitivity of 85–100% [17]. Another advantage is the understanding of the dynamics of T-cells in the skin and blood over time.

The method for skin clonality analysis is specified in the German guideline [18] and is currently based on the Biomed 2 protocol, using fragment length-based detection methods like gel-electrophoresis and gene scanning [15]. However, studies showed that high-throughput sequencing of the TCR genes (TRB and TRG) provides a superior tool for the diagnosis of CTCL and are of prognostic significance [19,20].

Next-generation high-throughput DNA sequencing of TRB has been validated in a cohort of 208 patients with CTCL in lesional skin. The tumor clone frequency (TCF) > 25% was an independent prognostic factor for both progression-free status and overall survival in patients with CTCL, and MF in particular [21]. The TCF is therefore an important factor for identifying patients at risk of progression, and the most important among the known prognostic factors, which include age of >60 years, elevated LDH, and large cell transformation in the skin [22]. TCR amplicons analyzed by accurate short read Illumina (NGS) sequencing allows for in-depth characterization and classification of the sample, based on individual clones identified through highly accurate sequences. Thus, sequencing methods in clinical daily practice would lead to an earlier and more precise diagnosis and might aid in the prognosis of individual patients [21,23]. However, even the significantly reduced sequencing costs [24] have not yet ensured the widespread use of NGS in a clinical setup, and NGS is currently associated with considerably high costs. Cost-effective sequencing can only be achieved if the instrument is fully utilised, but the demand for sequencing diagnostics in a clinic is not sufficient to achieve this level of efficiency. The high operating costs are split into maintenance contracts and fixed costs for the flow cells. Here, a lower sample amount is not compatible with lower sequencing costs, which is why the time difference until diagnosis would be too long if the sample cohort were to be kept economical [25]. Thus, there is a high need for highly accurate sequencing methods based on small inexpensive instruments, flexible sample throughput, and a short run time [26].

It is therefore important to develop alternative methods that are more cost-efficient and still reliable for the diagnosis of TCR rearrangement.

This study compares the sequencing accuracy of TRB and TRG repertoires from CTCL patients using NGS (Illumina MiSeq) and third generation (ONT GridION) sequencing. Whereas NGS is an established technology, showing very low error rates, third-generation sequencing technologies do have considerably higher error rates, with around 1% error in the current version of the nanopore sequencing technology. On the other hand, this technology is faster than NGS, has a smaller economic footprint, and is suited even for small labs [27,28].

The hypothesis of this study is that the sequencing accuracy using ONT is sufficient such that the most abundant TCR clones are found at similar frequencies as in the current “gold standard” NGS. Consequently, the results of TCR clonality analyses would not be significantly different from each other.

## 2. Materials and Methods

### 2.1. Sample Collection

In this study, material from 45 patients with various diagnoses was analyzed: Mycosis fungoides (37/45), Sézary Syndrome (2/45), folliculoptropic CTCL (1/45), and non-CTCL diagnoses as polyclonal controls (5/45) were included Table 1. These analyzes included three different sample types: formalin-fixed and paraffin-embedded tissue (FFPE) biopsies (27/45) (Appendix A), fresh frozen tissue samples (9/45) and isolated CD3+ cells from fresh punch biopsies (9/45) (Appendix A). A Jurkat cell line (E6-1) was used as a monoclonal control.

### 2.2. Sample Processing

For the fresh punch biopsies, CD3+ isolation was carried out [24]. From single cells as well as from fresh frozen samples, DNA was isolated using the DNeasy Blood & Tissue Kit (Qiagen, Hilden, Germany). For FFPE samples, DNA was isolated using the QIAamp DNA FFPE Tissue Kit (Qiagen, Hilden, Germany).

### 2.3. Amplicon Preparation

The general concept of the amplicon preparation was based on the publications from the EuroClonality–NGS working group from 2019 [29,30,31]. Modifications from the protocol are listed accordingly. If available, all samples were brought to a concentration of 20 ng/µL in a volume of 5 µL (in total 100 ng). If a lower DNA concentration was available, no dilution of the DNA was performed.

For each patient sample two multiplex PCR reactions, TRB-VJ (52 Primers) and TRG (12 Primers) were performed using the primer sequences and the PCR cycling protocol of the Standard Operating Procedure for the MiSeq platform from the EuroClonality–NGS working group [32]. One difference from the EuroClonality–NGS protocol is that no spike-in DNA and no buffy coat was used. Furthermore, the amplificants were then purified via AMPure XP Beads in a proportion of 1:1 instead of gel extraction, which was also tested by the EuroClonality–NGS working group as a good alternative [29].

In the initial phase, the fragment lengths of some of these samples were analyzed due to the purity of the expected DNA fragment length [32] by capillary electrophoresis (2100 Agilent Bioanalyzer). We confirmed the expected fragment lengths (TRG 160 bp; TRB-VJ 200–216 bp) in stitched samples. This served as the basis for proceeding without additional fragment length controls before advancing to the second PCR.

After QuBit 4 measurement, if available up to 10 ng purified DNA was used for the second PCR. Unique barcoding per sample was performed using Illumina TruSeq adapter sequences, which were already included in the forward primers, Index 2 (i5, D501–D508), and reverse primers with reverse-complementary Index 1 (i7, D701–712). The second PCR protocol was also performed according to the EuroClonality–NGS working group guidelines, followed by AMPure XP Bead clean up as previously described after the first PCR. To check the purity of the expected amplicon sizes of TRG (256–360 bp) and TRB-VJ (309–407 bp) and to quantify these fragment sizes of each sample, a measurement was conducted (2100 Agilent Bioanalyzer or Agilent Technologies and 5300 Fragment Analyzer System, Agilent Technologies, Santa Clara, CA, USA). Based on these results, libraries were then equimolar combined in two pools, TRG and TRB-VJ. Finally, the two pools underwent a precise fragment-based cleaned up using a BluePippin (Sage Science, Beverly, MA, USA, cassette 2%) based on the expected fragment sizes before equimolar pooling.

### 2.4. Sequencing of TCR Library and Data Analysis

For MiSeq sequencing (Reagent Kit v3, Sage Science, Beverly, MA, USA) 2 × 300 nt paired-end reads were generated. In parallel, the GridION sequencing platform (ONT) was used. Due to the rapid development of the ONT platforms and the associated bioinformatics, the samples could not all be processed in the same way. Fresh frozen samples were sequenced using R.10.4 flow cells in combination with the SQK-Q20 early access kit with the base calling option *super accuracy* (Guppy version 6.1.5). For the CD3-isolated cell library samples, R.10.4.1 flow cells in combination with SQK-LSK114 chemistry and Guppy version 6.3.9 were used. The library, including the FFPE samples, was ONT sequenced with the SQK-LSK114 kit, and base called with Guppy version 7.0.9 in super-accurate base calling mode by 400 bps with a minimum detected read length of 200 bp.

The ONT SQK-protocol was optimized in the cleanup step after adapter ligation using an adapted AMPure XP Bead amount of 90 µL instead of 40 µL, followed by washing with 250 µL short fragment buffer.

The generated single-end nanopore data and paired-end reads generated by Illumina were analyzed as described in the following. The demultiplexing pipeline that was used is designed for mixed patient data, which has been sequenced without prior ONT barcoding, but using Illumina adapters (iD5 and iD7) previously added in the second PCR. Each barcoded primer adapter could then be precisely assigned to a patient and the target (TRG, TRB-VJ).

The analysis of the demultiplexed *fastq* data in the clonotypes was performed using the tool MixCR with the standard parameters, including the commands for alignment, assembling, export of the alignments and clones, and data converting for VDJviz 1.0.4 [33]. The resulting .txt file was then uploaded to VDJviz for visualization of the clones and their frequencies [34]. The clone frequencies calculated by MixCR could then be used for TCF analysis to gain a better understanding of the profiling status of the sample. The mean number of rearranged TRG alleles for a total population of mature human T-cells is 1.8 per cell, which is due to the fact that most TCRs have two rearranged gamma alleles [35]. Thus, the TCF calculation of the TRG region was performed using the top two most frequent clone sequences (Formula (1)), while the TCF of the TRB chain was calculated using the frequency of only the first top clone sequence (Formula (2)).

Tumor clone frequency (TCF) was calculated by:(1)$$TCF\;TRG=\frac{f1+f2}{fi}$$
(2)$$TCF\;TRB=\frac{f1}{fi}$$

f1 = total reads of the most abundant TCR nucleotide sequence in the sample;

f2 = total reads of the second most abundant TCR nucleotide sequence in the sample;

fi = total reads of the sample.

For statistics and correlation illustrations, OriginPro, version 2021b (OriginLab Corporation, Northampton, MA, USA) was used.

## 3. Results and Discussion

We have developed a protocol for TCR analyses using nanopore sequencing. To demonstrate the validity of our protocol we compared it with the current gold standard, Illumina-based sequencing of amplicons according to the EuroClonality protocol. For these analyses we used CTCL patient samples of different types. As controls, we used polyclonal samples from patients with benign skin diseases and a monoclonal Jurkat cell line. As a first step, the TCR amplicon workflow of the EuroClonality consortium was adopted for ONT sequencing. This included the simplification of this protocol, e.g., without the use of spike-in DNA or excision from the agarose gel. The TRG and TRB amplicons generated in this way were halved and each analyzed by ONT and Illumina sequencing.

The sequencing data for the top TRB and TRG clones in the clonal samples showed identical sequences obtained by both ONT and NGS techniques (Illumina, Figure 1; tool used: Blastn, NCBI). In addition, the TCF values for TRG and TRB showed a high correlation between the two techniques. This correlation held true for the Jurkat cell line as well as for the FFPE, FF, and CD3+ isolated cell samples. Furthermore, the calculated TCF values from the Jurkat cell line showed consistent results between the two techniques and across replicates for TRG (TCF ONT: 92; 92 and TCF Illumina: 100; 100) and for TRB-VJ (TCF ONT: 96; 98 and TCF Illumina: 99; 99). The TCF values of the FFPE samples, calculated from the nanopore sequencing data, exhibited a high correlation with those from the NGS data (R = 0.994 for TRG; R = 0.99334 for TRB), as shown in Figure 1a,b. The high accuracy was also confirmed with the TCF values from the isolated CD3+ cells shown in Figure 1c,d (r = 0.993 for TRG; r = 0.994 for TRB) and for the fresh frozen samples in Figure 1e,f (r = 0.996 for TRG; r = 0.992 for TRB). The low TCF factor (<10) of some patient samples is due to polyclonal control samples originating from healthy patient samples, representing a broad range of different T-cell receptors. In this context, the percentage rearrangement of TCR β and TCR***γ*** receptors is well below a TCF of 25. Conclusions regarding initial clonality must always be weighed up in the context of the histology, immunohistology, and the clinical picture of the individual. Furthermore, the TCR receptors TRG and TRB are not always rearranged simultaneously.

To minimize sequencing errors related to depth, the theoretical minimum sequencing depth for analyzing Illumina and ONT data was set at 20,000 reads per sample. However, in practice, in this study a minimum depth of 45,800 reads was achieved.

The good correlation results show that sequencing by ONT may offer new opportunities; the advantage of ONT is the scalable run time, depending on the number of samples and the required sequencing depth, compared to the fixed run time of approximately 56 h without any time or cost reduction when using an Illumina MiSeq.

In addition, after sequencing, depending on the run time, there may be enough pores intact for another run, so the flow cell can be washed and reused. This enables the analysis of small sample volumes from critical clinical cases that require rapid results, as immediate treatment decisions must be made depending on the sequencing result. Low investment costs combined with low-cost sequencing runs emphasize the benefits of this technology.

An example cohort of 10 patients resulted in a cost of €238 per sample using Illumina sequencing. Cost reduction per sample with Illumina technology can only be achieved if additional samples are included in the run. In contrast, ONT sequencing allows for the adjustment of sequencing depth, providing flexibility. For instance, with ONT sequencing, the cost per sample would be €112 when processing 10 samples in parallel. The cost analysis includes both the simultaneously performed steps, such as DNA isolation, and all technology-specific adaptations, such as fragment-length analysis and sequencing itself. Additionally, the acquisition costs for ONT technology are in the lower four-figure range and include a starter kit for sequencing (€1999 for a MinION Mk1B with two flow cells, as of 25 October 2024). In comparison, an Illumina MiSeq is priced at approximately €1956.00 (as of 25 October 2024).

However, to date, the application of ONT has been hampered by a low raw read accuracy, and erroneous base calls on homopolymers [36,37]. Further development of this technology, consisting of constant adjustments to the chemistry, flow cells, and software, has resulted in an accuracy that has now risen to 99% [38], which can be supported by the data generated in this study.

Over time, ONT has also been used for various medical applications like antibiotic resistance gene identification, HLA haplotype variant detection, and Ebola surveillance [39,40,41]. The small ONT MinION device has already been successfully used for bacterial strain identification in places with challenging conditions such as space stations or on ships [25,42,43,44], demonstrating the low requirements for this technology.

The reduction of sequencing results to the most abundant clones could raise questions about the accuracy of all identified clones per sample. However, the similarity of the chord diagrams in Figure 2 indicates a strong resemblance in the total number of clones found per sample, as demonstrated for both a polyclonal and a monoclonal example. CDR3 similarity, based on the same TCR rearrangement events of the variable (V) and joining (J) allele in the sample, is shown for TRG in a monoclonal example (Figure 2a,b) and a polyclonal example (Figure 2c,d). Minimal percentage changes for the abundance of the clones in the polyclonal samples led to a change in the order of the diversity among the most represented receptor sequences. However, no significant change in the TCF values was observed.

Replicates of library preparation and subsequent sequencing demonstrate minimal or negligible variance in both MiSeq- and ONT-sequenced datasets (Figure 3). This consistency is observed across samples with a highly diverse TCR repertoire (healthy donor and PBMC sample) and samples exhibiting strong monoclonality (Jurkat cell line). Consequently, the stability of the measured TCF values is confirmed for both TRG–TCF and TRB–TCF values, supporting a degree of reliability in the ONT-derived results. A noteworthy observation emerged when an ONT sample was analyzed using an updated Guppy version; error rates decreased with each successive platform version (Guppy version 5: TCF 87, 88; Guppy version 6: TCF 94). Therefore, it is recommended that bioinformatic analyses employ the latest Guppy version for base calling (at least version 6.1.5) as older versions may significantly impact base detection accuracy.

## 4. Conclusions

In this study we have demonstrated the feasibility of TCR clonality analysis using ONT sequencing. Our results revealed a robust correlation in TCF values between data generated by ONT sequencing and the current gold standard NGS method (Illumina). This was verified using Jurkat cell line replicates and 45 patient samples of different sample types: fresh frozen, formalin-fixed and paraffin-embedded, and single cell suspensions of CD3+ cells. The correlated values for the tumor frequency factor (TCF) were calculated from the frequency of the top clone(s) found in the sample. In conclusion, the results described here offer the possibility of using a simpler, faster, and cheaper sequencing technique for both TRG and TRB chains in a more individualized smaller sample setting. However, for large sample sizes (PCR-based and PCR-free projects) Illumina may be the more cost-effective alternative.

Routine implementation of this protocol would require verification and validation of the amplicon generation in conjunction with the ONT sequencing platform to ensure accuracy in compliance with in vitro diagnostic guidelines. Widespread adoption of this technology in diagnostic approaches could unlock the potential to enable earlier detection and monitoring procedures for smaller patient cohorts, such as CTCL patients, ultimately improving the prospects of early and successful treatment.

## Figures and Tables

**Figure 1 cancers-16-03700-f001:**
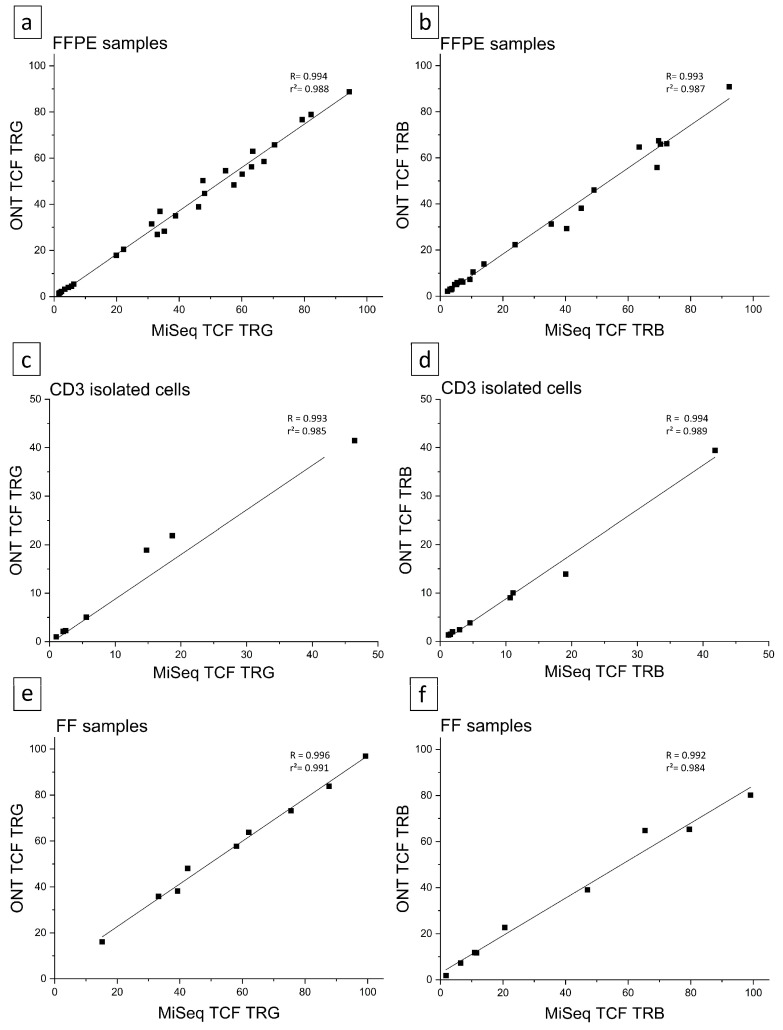
Correlation of TCF data for TCR analysis via Illumina (MiSeq) or ONT amplicon sequences. For all samples the calculated TCF values for both analyzed T-cell receptor chains (TRG and TRB) are shown and correlation between both sequencing methods is depicted. (**a**,**b**) FFPE samples, TRG *n* = 27, TRB *n* = 23. (**c**,**d**) CD3-isolated cell samples, TRG *n* = 9, TRB *n* = 9. (**e**,**f**) fresh frozen (FF) samples, TRG *n* = 9, TRB *n* = 9.

**Figure 2 cancers-16-03700-f002:**
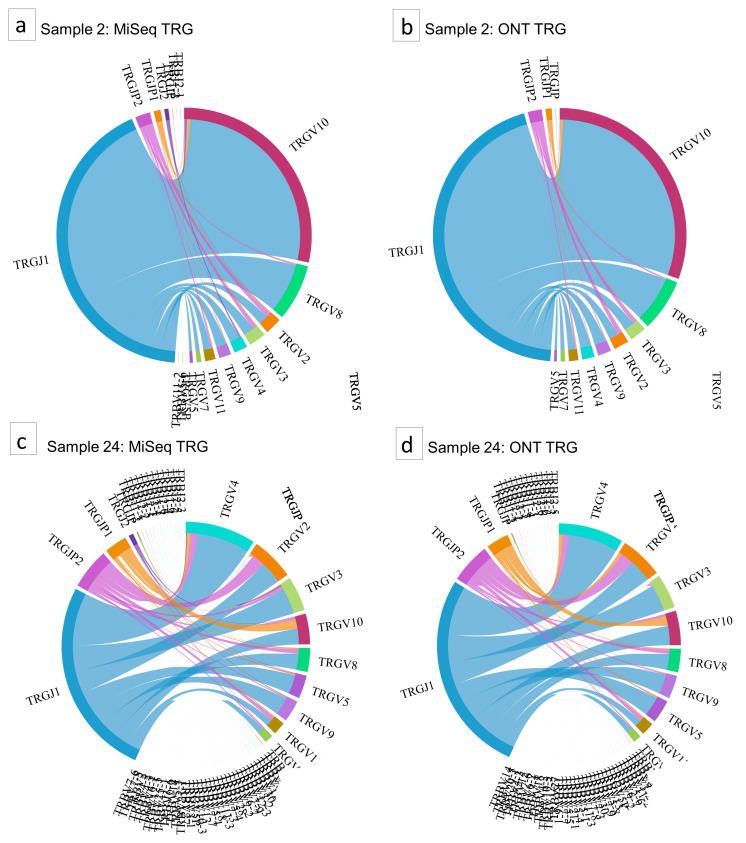
Chord diagrams of TRG repertoires. A monoclonal sample in TRG sequenced by MiSeq (**a**) and ONT (**b**) compared to a polyclonal example in TRG analyzed with MiSeq (**c**) and ONT (**d**).

**Figure 3 cancers-16-03700-f003:**
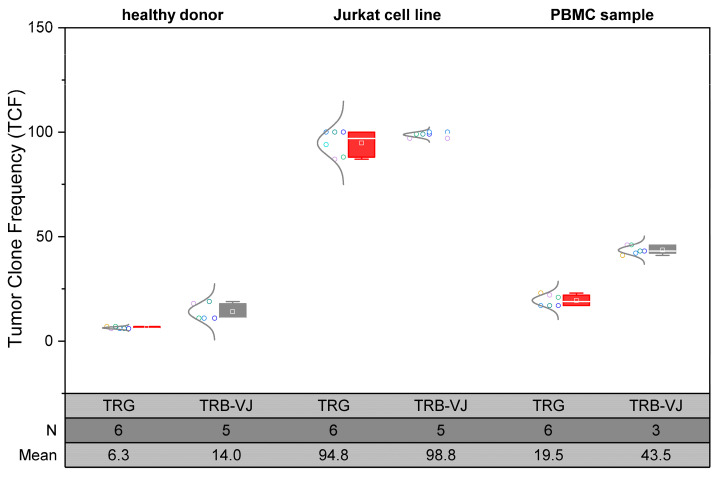
Relative quantification of the top clones (TCF) in replicates for 3 samples: a healthy donor, Jurkat cell line and a PBMC sample isolated from a patient with active immune state. The blue spots are the MiSeq (Illumina) data; the green spots, the ONT-Q20 data (Guppy Version 5.1.13); and the one spot in purple (in TRG–Jurkat 100%) is another ONT-Q20 spot analyzed with Guppy version 6.1.5.

**Table 1 cancers-16-03700-t001:** Characteristics of patient cohort.

Characteristics			
Age	median	64	
	mean	65.8	
	range	26–92	
Sex	female	13 (28.9%)	
	male	32 (71.1%)	
**Diagnosis**	**FFPE** **(*n* = 27)**	**CD3 cells** **(*n* = 9)**	**Fresh Frozen** **(*n* = 9)**
Mycosis fungoides	19 (70.4%)	9 (100.0%)	9 (100.0%)
Sezary Syndrom	2 (7.4%)		
Other Cutaneous T-cell Lymphoma	1 (3.7%)		
Other benign skin diseases	5 (18.5%)		
**EORTC stage ***	**FFPE** **(*n* = 27)**	**CD3 cells** **(*n* = 9)**	**Fresh Frozen** **(*n* = 9)**
IA	10 (37.0%)	7 (77.8%)	
IB	4 (14.8%)	1 (11.1%)	2 (22.2%)
IIB	5 (18.5%)	1 (11.1%)	7 (77.8%)
Erythroderma, IVA1	2 (7.4%)		
IVA2	1 (3.7%)		
no tumor	5 (18.5%)		

* [18].

## Data Availability

The data can be shared on request.

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
