# Peer review of "Nanopore Sequencing for T-Cell Receptor Rearrangement Analysis in Cutaneous T-Cell Lymphoma"

_cancers, 2024, doi:10.3390/cancers16213700_

Round 1
Reviewer 1 Report
Comments and Suggestions for Authors
The manuscript describes a method for analyzing T-cell receptor (TCR) rearrangements in patients with cutaneous T-cell lymphoma (CTCL) using ONT nanopore sequencing technology, particularly in the diagnosis of CTCL such as Mycosis fungoides (MF) and Sézary syndrome (SS). The study demonstrates the efficiency and potential of nanopore sequencing technology by comparing it with the existing Illumina sequencing technology, which is innovative.
Major:
1. The authors note that Illumina (NGS) sequencing is not widely used for analyzing T-cell receptor (TCR) clonality due to certain limitations and ONT sequencing maybe potential as a faster and more cost-effective option. However, the manuscript does not provide theoretical costs to support this claim.
2. In Figure 1, many of the samples have TCF TRG or TRB scores below 5, which presents a puzzling distribution. Therefore, the authors should include more information about the sequencing depth for both NGS and ONT sequencing.
3. In Figure 2, the text in the image is overlapped, making it difficult to read.
4. Line 49, 294 appears to have a formatting error.
5.The presentation of graphs and data can further refine the detailed statistical analysis of different samples and provide readers with more information about the distribution of sequencing results. In addition, repeated experimental data of different samples can be added to further enhance the credibility of the experimental results.
Author Response
Major:
- The authors note that Illumina (NGS) sequencing is not widely used for analyzing T-cell receptor (TCR) clonality due to certain limitations and ONT sequencing maybe potential as a faster and more cost-effective option. However, the manuscript does not provide theoretical costs to support this claim.
The Illumina costs vary depending on the number of samples. For example, for 6 samples, the cost is €319, and for 22 samples, the cost is €180.
In contrast to Illumina, single samples can be analysed with ONT. The cost per sample is
An example cohort of 10 patients, resulting in a cost of €238 per sample using Illumina sequencing. Cost reduction per sample with Illumina technology can only be achieved if additional samples are included in the run. In contrast, ONT sequencing allows for adjustment of sequencing depth, providing flexibility. For instance, with ONT sequencing, the cost per sample would be €112 when processing 10 samples in parallel. The cost analysis includes both the simultaneously performed steps, such as DNA isolation, and all technology-specific adaptations, such as fragment length analysis and sequencing itself. Additionally, the acquisition costs for ONT technology are in the lower four-figure range and include a starter kit for sequencing (€1,999 for a MinION Mk1B with two flow cells, as of October 25, 2024). In comparison, an Illumina MiSeq is priced at approximately €1,956.00 (as of October 25, 2024).
2.
- In Figure 1, many of the samples have TCF TRG or TRB scores below 5, which presents a puzzling distribution. Therefore, the authors should include more information about the sequencing depth for both NGS and ONT sequencing.
The low TCF factor is due to polyclonal control samples originating from healthy patient samples, representing a broad range of different T-cell receptors. In this context, the percentage rearrangement of TCR β and TCR? receptors is well below a TCF of 25. Furthermore, the TCR receptors TRG and TRB are not always rearranged simultaneously. Conclusions regarding initial clonality must always be weighed up in the context of the histology, immunohistology and clinical picture of the individual.
To minimize sequencing errors related to depth, the theoretical minimum sequencing depth for analyzing Illumina and ONT data was set at 20,000 reads per sample. However, in practice, a minimum depth of 45,800 reads was achieved.
- In Figure 2, the text in the image is overlapped, making it difficult to read.
Corrected
- Line 49, 294 appears to have a formatting error.
Corrected
- The presentation of graphs and data can further refine the detailed statistical analysis of different samples and provide readers with more information about the distribution of sequencing results. In addition, repeated experimental data of different samples can be added to further enhance the credibility of the experimental results.
Relative quantification of the top clones (TCF) measured in replicates for 3 samples:
Replicates of library preparation and subsequent sequencing demonstrate minimal or negligible variance in both MiSeq- and ONT-sequenced datasets (Figure 3). This consistency is observed across samples with a highly diverse TCR repertoire (patient healthy and PBMC sample) and samples exhibiting strong monoclonality (Jurkat cell line). Consequently, the stability of measured TCF values is confirmed for both TRG-TCF and TRB-TCF values, supporting a degree of reliability in the ONT-derived results. A noteworthy observation emerged when an ONT sample was analyzed using an updated Guppy version; error rates decreased with each successive platform version (Guppy version 5: TCF 87, 88; Guppy version 6: TCF 94). Therefore, it is recommended that bioinformatic analyses employ the latest Guppy version for basecalling, at least version 6.1.5, as older versions may significantly impact base detection accuracy.
Reviewer 2 Report
Comments and Suggestions for Authors
The authors presented a workflow in this manuscript for analyzing the T-cell receptor repertoire based on third-generation nanopore sequencing, which offers reduced time consumption and cost-effectiveness in comparison to NGS-based techniques. This study aimed to emphasize the importance of establishing a fast and cost-effective procedure in T-cell receptor clonality analysis, which is of significant importance for improving diagnostic strategies for cutaneous T-cell lymphomas. The work is attractive and holds certain clinical significance but it should be modified on some points:
1. In the “Materials and Methods 2.3. Amplicon preparation” section, the authors have provided a detailed description of the amplicon preparation method used in this study, which is based on the standard operating procedure of the EuroClonality-NGS working group with some modifications. However, it is necessary to provide a clearer explanation of how these modifications might impact the experimental results and reproducibility. For instance, when mentioning the use of "AMPure XP Beads" (Pg4 line 137) as a purification method, it is important to provide more detailed specifications for this tool. Furthermore, in the last step of using BluePippin (Pg4 line 155) for fragment-based cleanup, it is recommended to provide more information about the operational details and procedures to ensure the generation of high-quality sequencing libraries.
2. In the “Results and Discussion” section (lines 197 to 232), the authors compared the applications of ONT sequencing and Illumina sequencing in TCR analysis, demonstrating their consistency and correlation. The study provided a clear description of the experimental procedures and results, supported by relevant figures. The authors also mentioned the advantages of ONT sequencing, such as adjustable run time and low investment costs, which are important for clinical cases requiring rapid results. However, this study could be further improved by discussing the strengths and limitations of ONT sequencing. While the authors mentioned that the accuracy of ONT sequencing has been improved to 99% (Pg6 line 231), the potential error rates and challenges in data interpretation in specific circumstances are supposed to be discussed as well.
3. In the “Results and Discussion” section (lines 244 to 253), the authors briefly described the CDR3 similarity based on TCR rearrangement events, and mentioned the minimal percentage changes in clone abundance in poly-clonal samples. However, the impact of these changes on the TCF values should be elaborated upon. The authors should provide more analysis and interpretation of the results, including statistical significance when applicable.
In the “Results and Discussion” section (lines 244 to 253), the authors briefly mentioned the importance of using the latest version of Guppy for basecalling. However, this statement lacked supporting evidence or data to substantiate its importance. The authors should provide more details and possibly include relevant results demonstrating the impact of Guppy version on base detection.
Author Response
The manuscript describes a method for analyzing T-cell receptor (TCR) rearrangements in patients with cutaneous T-cell lymphoma (CTCL) using ONT nanopore sequencing technology, particularly in the diagnosis of CTCL such as Mycosis fungoides (MF) and Sézary syndrome (SS). The study demonstrates the efficiency and potential of nanopore sequencing technology by comparing it with the existing Illumina sequencing technology, which is innovative.
Major:
- The authors note that Illumina (NGS) sequencing is not widely used for analyzing T-cell receptor (TCR) clonality due to certain limitations and ONT sequencing maybe potential as a faster and more cost-effective option. However, the manuscript does not provide theoretical costs to support this claim.
The Illumina costs vary depending on the number of samples. For example, for 6 samples, the cost is €319, and for 22 samples, the cost is €180.
In contrast to Illumina, single samples can be analysed with ONT. The cost per sample is
An example cohort of 10 patients, resulting in a cost of €238 per sample using Illumina sequencing. Cost reduction per sample with Illumina technology can only be achieved if additional samples are included in the run. In contrast, ONT sequencing allows for adjustment of sequencing depth, providing flexibility. For instance, with ONT sequencing, the cost per sample would be €112 when processing 10 samples in parallel. The cost analysis includes both the simultaneously performed steps, such as DNA isolation, and all technology-specific adaptations, such as fragment length analysis and sequencing itself. Additionally, the acquisition costs for ONT technology are in the lower four-figure range and include a starter kit for sequencing (€1,999 for a MinION Mk1B with two flow cells, as of October 25, 2024). In comparison, an Illumina MiSeq is priced at approximately €1,956.00 (as of October 25, 2024).
- In Figure 1, many of the samples have TCF TRG or TRB scores below 5, which presents a puzzling distribution. Therefore, the authors should include more information about the sequencing depth for both NGS and ONT sequencing.
The low TCF factor is due to polyclonal control samples originating from healthy patient samples, representing a broad range of different T-cell receptors. In this context, the percentage rearrangement of TCR β and TCR? receptors is well below a TCF of 25. Furthermore, the TCR receptors TRG and TRB are not always rearranged simultaneously. Conclusions regarding initial clonality must always be weighed up in the context of the histology, immunohistology and clinical picture of the individual.
To minimize sequencing errors related to depth, the theoretical minimum sequencing depth for analyzing Illumina and ONT data was set at 20,000 reads per sample. However, in practice, a minimum depth of 45,800 reads was achieved.
- In Figure 2, the text in the image is overlapped, making it difficult to read.
Corrected
- Line 49, 294 appears to have a formatting error.
Corrected
- The presentation of graphs and data can further refine the detailed statistical analysis of different samples and provide readers with more information about the distribution of sequencing results. In addition, repeated experimental data of different samples can be added to further enhance the credibility of the experimental results.
Relative quantification of the top clones (TCF) measured in replicates for 3 samples:
Replicates of library preparation and subsequent sequencing demonstrate minimal or negligible variance in both MiSeq- and ONT-sequenced datasets (Figure 3). This consistency is observed across samples with a highly diverse TCR repertoire (patient healthy and PBMC sample) and samples exhibiting strong monoclonality (Jurkat cell line). Consequently, the stability of measured TCF values is confirmed for both TRG-TCF and TRB-TCF values, supporting a degree of reliability in the ONT-derived results. A noteworthy observation emerged when an ONT sample was analyzed using an updated Guppy version; error rates decreased with each successive platform version (Guppy version 5: TCF 87, 88; Guppy version 6: TCF 94). Therefore, it is recommended that bioinformatic analyses employ the latest Guppy version for basecalling, at least version 6.1.5, as older versions may significantly impact base detection accuracy.
Figure 3: Relative quantification of the top clones (TCF) in replicates for 3 samples: a healthy donor, Jurkat cell line and a PBMC sample isolated from a patient with active immun state. The blue spots are the MiSeq (Illumina) data, the green spots, the ONT-Q20 data (Guppy Version 5.1.13) and the one spot in purple (in TRG-Jurkat 100%) is another ONT-Q20 spot analysed with Guppy Version 6.1.5.
Reviewer 3 Report
Comments and Suggestions for Authors
The study of Cieslak et al. is well-designed and the results are interesting. It would be wise to provide the reader with the real cost per case by adopting the two approaches. In fact, one of the limitations of NGS on the MiSeq platform is the need to collect several samples in order to reduce the test cost, this also affecting the turn-around time of the diagnostic process.
Author Response

(The authors gave the same response as above.)

Round 2
Reviewer 1 Report
Comments and Suggestions for Authors
The script is well written and I have no further comments.